# “*In the past, the seeds I planted often didn’t grow*.” A Mixed-Methods Feasibility Assessment of Integrating Agriculture and Nutrition Behaviour Change Interventions with Cash Transfers in Rural Bangladesh

**DOI:** 10.3390/ijerph17114153

**Published:** 2020-06-10

**Authors:** Ashraful Alam, Wajiha Khatun, Mansura Khanam, Gulshan Ara, Anowarul Bokshi, Mu Li, Michael J. Dibley

**Affiliations:** 1School of Public Health, The University of Sydney, Edward Ford Building, Sydney 2006, Australia; wajiha.khatun@gmail.com (W.K.); mu.li@sydney.edu.au (M.L.); michael.dibley@sydney.edu.au (M.J.D.); 2International Centre for Diarrhoeal Disease Research, Dhaka 1212, Bangladesh; mansura@icddrb.org (M.K.); gulshan.ara@icddrb.org (G.A.); 3School of Life and Environmental Sciences, The University of Sydney, Sydney 2006, Australia; anowarul.bokshi@sydney.edu.au

**Keywords:** nutrition-sensitive agriculture, nutrition behaviour change, social safety net, feasibility study, mixed-methods, mHealth, Bangladesh

## Abstract

Combining agriculture with behaviour change communication and other nutrition-sensitive interventions could improve feeding practices to reduce maternal and child undernutrition. Such integrated intervention requires rigorous design and an appropriate implementation strategy to generate an impact. We assessed feasibility and acceptability of an intervention package that combines nutrition counselling, counselling and support for home-gardening, and unconditional cash transfers delivered to women on a mobile platform for improving maternal and child nutrition behaviours among low-income families in rural Bangladesh. We used mixed-methods including in-depth interviews with women (20), key-informant interviews with project workers (6), and a cross sectional survey of women (60). Women well-accepted the intervention and reported to be benefited by acquiring new skills and information on home gardening and nutrition. They established homestead gardens of seasonal vegetables successfully and were able to find a solution for major challenges. All women received the cash transfer. Ninety-one percent of women spent the cash for buying foods, 20% spent it on purchasing seeds or fertilizers and 57% used it for medical and livelihood purchases. Project staff and mobile banking agent reported no difficulty in cash transfer. Combining nutrition-specific and -sensitive interventions is a feasible and acceptable approach. Using mobile technologies can provide additional benefits for the intervention to reach the disadvantage families in rural settings.

## 1. Introduction

Despite remarkable progress in agricultural sectors in Bangladesh, food and nutrition insecurity among the poor is common. Approximately 40 million people, one quarter of total population, are food insecure [1] and a nearly similar proportion of the population, one in four people, live in poverty [2,3]. The level of undernutrition in Bangladesh, particularly in women and children, is one of the highest in the world: 33% of children under five years of age are underweight and 36% are stunted; nearly one in five of the women suffer from chronic energy deficiency [4]. Over the past two decades domestic production of grains has significantly increased, yet crop diversification remains limited [5,6]. Poverty, low income and price hikes restrict the poor from accessing sufficient and diversified nutritious foods, which leads to their food and nutrition insecurity [7].

To generate positive impacts on nutritional outcomes, agricultural interventions require more focus on nutrition, and need to be linked to nutrition specific interventions [8]. Improving homestead fruit and vegetable production integrated with enhanced communications about nutrition will lead to improved dietary diversity and nutritional status of women and children [9,10]. Moreover, empowering women through their involvement in agricultural activities is crucial to effectively improve household food security, dietary diversity and child nutrition in South Asia including Bangladesh [11,12,13]. A recent systematic review revealed that unconditional cash transfer allowed the poor households to secure access and consumption of a variety of foods in low- and middle-income countries (LMIC) [14]. Hence the 2015 Food and Agriculture Organisation (FAO) Report on “The State of Food and Agriculture” suggests that the social protection be linked with agriculture in order to eradicate rural poverty, food insecurity and hunger, which is essential to improve maternal and child nutrition among the poor [15].

Currently in Bangladesh there are several parallel interventions in agriculture, nutrition and social protection that lack integration. Likewise, nutrition elements in agricultural interventions are largely non-existent. The Government of Bangladesh (GoB) has set 12 priority investment programmes to improve food security and nutrition in integrated ways, including sustainable and diversified agriculture through integrated research and extension, and community-based nutrition programmes and services. Although an integrated package of interventions across sectors is needed to create a more nutritious food system in Bangladesh, how to achieve this is less clear. 

Earlier evidence suggested that using mobile phones in agriculture extension could be a cost- effective tool for farmers to access information, contribute to management of input and output of the supply chain, and increase the accountability of extension activities [16,17]. Similarly, with maternal and child health and nutrition programmes, mobile technologies are increasingly being harnessed to improve the effectiveness of nutrition behaviour change communications in LMICs [18,19]. The widespread mobile network and rapidly expanding mobile phone ownership in Bangladesh provides an ideal opportunity to improve the dissemination of agricultural and nutrition information and cash transfer through mobile phone technologies [20,21].

We envisaged that combining agriculture and nutrition counselling with unconditional cash transfer using a mobile phone platform would be an effective strategy to improve maternal and child nutrition in the poor households, and with the potential for scale up in resource poor settings including Bangladesh. This study aims to test the feasibility, acceptability and the compliance of our intervention package in a poor rural community in Bangladesh. In addition, we wanted to assess the participant’s understanding, knowledge retention and perceived usefulness of both direct counselling and cash transfer using a mobile platform. Finally, we wished to explore the expenditure pattern of the cash transfer to see how the families utilised the money and to what extent it was spent on food, health needs, and livelihoods investments.

## 2. Materials and Methods

### 2.1. Study Design

We conducted a mixed-methods feasibility study. The study was conducted in two villages, Kurigram District in northern Bangladesh. We consulted the World Food Programme’s (WFP) Bangladesh Poverty Map [22] and the experts from the Bangladesh Agriculture Extension Programme (BAEP) to select the study district as one of the most impoverished and food insecure areas in the country. One village close to the Tista River and one further away were selected. One village was a little away from the river but had similar socio-economic conditions. Both sites were in Kurigram with about one kilometer apart. We listed and mapped the households in the study villages and conducted rapid household screening to identify the poorer households with women of reproductive age and under-five children. A total of 60 eligible households that consented were enrolled in August 2017 using a mobile phone-based registration system. The enrolled households received the intervention package for six months. 

### 2.2. Project Implementation

#### 2.2.1. Phase 1: Design of Intervention in Consultation with Stakeholders

We have undertaken this process in three steps as below.

We conducted a formative research in early 2017 to explore socio-cultural aspects of the study community to generate insights to develop the study design. The formative research involved in-depth qualitative interviews, key informant interviews and focus groups of women, their mothers-in-law, and the government agriculture extension workers from the study area. 

To finalise the design, in the first step, three members (Alam, Dibley and Bokshi) of the University of Sydney research team consulted the consortium members at the International Centre for Diarrhoeal Disease Research, Bangladesh (icddr,b) and Solidarity, the local implementation partner. In this meeting, the investigators reviewed the formative research findings to develop a draft study design including a description of the intervention package.

In the second step, the Solidarity team organised a stakeholder workshop in Kurigram which included stakeholders from the Kurigarm units of the Bangladesh Agriculture Research Institute (BARI) and BAEP, different NGOs, development, nutrition and human rights organisations, and the Civil Surgeon (chief health officer) of Kurigarm. Alam, Dibley and Bokshi presented the key results of the formative research. The stakeholders provided their feedback in an interactive workshop on the design and implementation strategy of the study. 

In the third step we finalised the design, the implementation plan, the training curriculum and materials, the communication tools, and evaluation plan based on the findings of the formative research and the stakeholder meeting. The formative research provided us insight into the key elements to be included in the educational messages. The nutrition and agriculture experts among the authors developed the messages that were shared with professionals from public and NGO sectors for feedback. We field-tested the final messages to check for understandability among the beneficiary community. A skilled programmer experienced in CommCare mobile phone applications, developed an app about these nutrition and agriculture messages that was embedded in the smart phones to be used by the trained counsellors for counselling the participating women. The app also included a resource library that accumulated pictorial illustrations and videos about appropriate infant feeding and home gardening practices. The counsellors showed the video or illustration linked to a specific message if a woman asked for an elaboration of the message. We tested and debugged the apps prior to their use in the field.

#### 2.2.2. Phase 2: Description of the Intervention

We developed a multi-component, community-based, intervention that combined agricultural and nutrition related activities. None of the women registered to the study owned a mobile phone. We provided a mobile phone and recharge to every woman who participated in the intervention to link the low-income families with agricultural extension and community nutrition programs, and to provide cash transfer for their participation. The women retained the phone at the end of the study. Our intervention package consisted of five components:*Registration and training of beneficiaries*: A project community facilitator contacted the eligible families, obtained their consent for participation, registered the 60 women of reproductive age with an under-five child, provided the women with a low cost mobile phone with the cash transfer application embedded, and trained them on the use of the mobile phone for receiving the money. The Table 1 describes the types of mobile phones and the recipients of the phones.*Information and support for more effective agricultural activities*: We provided the project agriculture workers based at Solidarity with a low-cost smartphone with an embedded application that included messages on homestead gardening and related pictorial and narrative materials. They were trained to use this mobile phone application to support their communications with the women and family members about appropriate homestead gardening practices, selection of crops, and nutritional aspects of the crops. The app also allowed them to gather basic information about their clients, monitor their activities and improve the supervision of their work.*Support for homestead gardening*: The trained government agriculture workers visited the households to assess the potential for developing homestead gardening, and provided the registered women and her husband with training on the usefulness of homestead gardens and specific skills such as preparation of garden beds, selecting appropriate seeds, when to plant, how to fertilize and irrigate the plot, harvesting the produce, and explanations as to where cash (see number 5) could be used to purchase inputs such as seeds and fertilizers.*Nutrition education and counselling*: Trained nutrition counsellors visited the home of the women fortnightly to counsel them and their husbands. Using the smartphone app with embedded text, videos and pictorial messages to support their nutrition counselling, the counsellors provided messages about appropriate diet for pregnant and lactating women and children. The counsellors used the app also to gather basic nutrition related information from the family, especially about the women and their children. Apart from the household visits, group counselling was provided to the women, their husbands and mothers-in-law in the first month of the intervention.*Cash transfer:* The participation of the family in the counselling activities was recorded. Each registered woman opened a bKash (the largest mobile banking operated in Bangladesh) account and received a monthly cash of Bangladesh Taka (BDT) 1200 (GBP 10) for six months. A text message was sent to the women’s mobile phone indicating the monthly cash was ready to draw from the designated bKash agent in the nearly local market. The woman or any family member could collect the money upon producing the message before the agent.

#### 2.2.3. Phase 3: Piloting of the Intervention

We assessed the feasibility and acceptability of our integrated agriculture-nutrition education and counselling and unconditional cash transfer intervention in a pilot study. 

### 2.3. Evaluation

#### 2.3.1. Qualitative Data Collection

We used in-depth interviews and key-informant interviews to gather the qualitative data. We conducted in-depth interviews of a randomly selected twenty women who had under-five children among the registered households. We explored their experience of participating in the intervention. Key-informant interviews were conducted with two nutrition counsellors, two agriculture counsellors, the project officer, and the Agriculture Extension Worker from the Government of Bangladesh Department of Agriculture Extension assigned in the area. These interviews generated detailed data on the implementation of the intervention including day to day operations, and the challenges and facilitators encountered as the project was rolled out. We developed a data collection guideline for each method, translated in Bengali, and pre-tested it in the study area. The guidelines were flexible to allow the necessary modifications to adjust any feedback derived from the field throughout the data collection. We captured the qualitative interviews on digital audio.

The study was approved by the Ethical Review Committee of the International Centre for Diarrhoeal Disease Research, Bangladesh (icddr,b). We obtained written informed consent from each participant of the survey, in-interviews and key-informant interviews. Verbal audio recorded consent was also obtained for audio recording of the interviews. The participants were assured that their participation in the data collection was completely voluntary. We also assured anonymity and privacy of information they would provide.

#### 2.3.2. Quantitative Data Collection

We conducted a survey at the end of intervention with a total of 58 out of 60 women who participated in the intervention. Information on household demography, participation in the counselling and homestead gardening, uses of mobile phones, utilisation of the cash, and issues around collecting money from the bKash agent was collected in the structured interviews. 

### 2.4. Data Analysis

*Qualitative*: The interviewers transcribed the audio-recorded interviews verbatim in Bengali and entered as Microsoft Word files. As the first step of coding, a qualitative researcher (Khatun) who speaks Bengali read a subset of the transcripts to identify the texts associated with the topics of research interest. These were assigned codes and verified by the senior health social scientist (Alam) for consistency and prepared the code list. We then applied the codes to all transcripts, compiled the data, and generated separate files with the text pertaining to each topical codes. Two authors (Alam and Khatun) read and reread the compiled data files separately to find themes and trends relating to the study objectives that were discussed with other Bangla speaking authors (Khanam and Ara). The broad themes generated from the data included the women’s level of understanding and satisfaction with the counselling, compliance towards the messages, barriers to establishing homestead garden, women’s mobile phone usage, their experiences on cash transfer through bKash, and spending of and decision making about the cash received. 

*Quantitative*: We performed descriptive statistical analysis to generate frequency tables using SPSS version 21. The quantitative data complemented the qualitative information. We assessed the participants’ socio-demographic characteristics, participation in both nutrition and home gardening counselling, types of information received, receipt of cash transfers, use of the cash, and use of mobile phone by women, in the qualitative analysis to complement the qualitative data.

## 3. Results

### 3.1. Qualitative Findings 

#### 3.1.1. Understanding the Project and its Implementation

The majority of the women stated that they were informed and understood the project’s aim, selection criteria and proposed interventions. To receive this information, they attended a courtyard meeting with a group of all study participants at the start of the intervention. They received a mobile phone to open a bKash account to receive the cash. All of them attended the agriculture and nutrition counselling session fortnightly. The counsellors also visited them at home weekly to monitor and supervise their homestead gardening activities. They also reported that the agriculture counsellor demonstrated the steps for establishing a homestead garden to a small group at one of their homesteads. 

#### 3.1.2. Women’s Participation and Related Factors

All participating women consented to take part in the study. However, some of them thought that family awareness was an important favourable factor for their participation in this study. Family and community members’ awareness and positive attitudes helped the women feel comfortable to be involved in this study. Women in general remained satisfied with their time allocated for counselling and homestead gardening.


*“Our husbands went with us (in the meeting). We, ourselves, shouldn’t only be aware (about the project activity), our husbands too need to be aware. Husbands usually take actions.”*
(Woman aged 20)

In response to the question about their time of involvement in counselling, the women mentioned that they had no problem finding time to participate. The time of the counselling was midmorning before lunch when women were usually free after completing their household chores in the morning. Their understanding on the importance of counselling motivated them to manage their time to take part in the counselling.


*“We can give time to the counsellors. They usually visited us at home when we had less household works. Sometimes they informed us before coming, sometimes did not.”*
(Women aged 19)

#### 3.1.3. Knowledge Retention from Agriculture Counselling

The information provided through agriculture counselling was well understood by the participating women. Most of them felt that the agriculture worker’s demonstration of gardening helped them learn about the practical steps of gardening such as soil and bed preparation.


*“Krishikorm’ (Agriculture Counsellor) told us to use a bucket for watering, use fertilizer or cow dung and pesticides. They told us to mix fertilizer with water and spray it.”*
(Woman aged 20)


*“They (Agriculture Counsellor) advised us to plough the land, apply cow dung, faeces of chicken and goat etc, and urea and patash (potassium)*
*fertilizers to make a bed to plant red amaranth, napa leaves and spinach. They asked us to make separate beds; if we make beds then excess water will drain. …To grow bottle gourd, they told us to dig [soil] one arm deep, apply cow dung and goat manure, keep it like that for 5/6 days and then plant gourd seeds.”*
(Woman aged 25)

#### 3.1.4. Compliance and Barriers to Establish Homestead Gardens

All participating women reported establishing their own homestead garden without much difficulty. Although some of them had either no experience of gardening or had failed to grow vegetables before the counselling, they succeeded after they were trained and counselled. Moreover, many of the women stated that the agriculture counsellor’s advice, regular monitoring, and supervision helped them to establish their homestead garden.


*“In the past, quite often the seeds I planted did not grow. I have been benefited after I planted according to the instructions of the ‘Krishikormi Dada’ (brother agriculture worker) about how to plant (seeds).”*
(Woman aged 32)

Some of the women faced a few barriers, in particular, with growing vegetables. The barriers included poor quality seeds, excessive monsoon rainfall, damage of the seed beds by hens and ducks, and destruction of the garden by goats and cows. The families fenced the garden with bamboo sticks to protect their gardens from the predators. The counselling and cash transfer enabled them to identify and purchase good seeds from the nearby market.


*“Rainfall washed away (the seeds) but I planted the seeds again. Cow and goats have damaged the vegetables, so I have surrounded (the garden with fence) as advised by the Krishikormi Bhai (brother Agriculture Counsellor).”*
(Woman aged 20)

#### 3.1.5. Understanding the Nutrition Messages

Almost all of the women had no difficulties to understand the messages delivered by the nutrition workers through direct counselling. They reported receiving new information on appropriate infant feeding including breastfeeding practices, frequency of infant feeding, consumption of a variety of foods including vegetables and animal source foods for better nutrition of mothers and their children. The videos in the counselling app were clear and easy for them to understand.


*“It was in the video that a mother took rice, vegetables and egg in a bowl of a quarter of a litre. The video showed, when the child eats, the mother talks with the baby to introduce the curry (to the baby). When the mother went away for other works, child stopped eating! It also displayed hand washing of a mother and her child using soap, cleaning of plates before eating, and positioning of two fingers (around the nipple) while breastfeeding.”*
(Woman aged 20)

#### 3.1.6. Perceptions on Benefits of Nutrition Sensitive Agriculture Counselling

The women generally believed that the agriculture counselling helped them learn new techniques on homestead gardening. In addition, nutrition counselling created awareness to consume the vegetables that they produced themselves for better nutrition and health of them and their children.


*“Counselling seems to be good for me. I liked it because I learned from it. I have learned about vegetable gardening and how to feed the nutritious foods to my children. I also liked the nutrition information as I didn’t know such information earlier. Now I know and understand importance of eating nutritious food; (now I know) feeding (the child) vegetables would provide vitamins.”*
(Woman aged 22)

#### 3.1.7. Women’s Perceptions on Mobile Phone Ownership and Usage

Almost all of the women did not have their own mobile phones, and they were pleased to receive a mobile phone from the project. They mostly used the phone for calling but some of them also used it for listening to music. However, only a few of them could read or write an SMS. Usually they asked for help from a family member who was literate and familiar with the SMS function of mobile phone. Charging the phone often appeared as a barrier to use the phone.

The ownership of a mobile phone benefited the women in communicating easily with other family members, relatives and the counsellors. They could use this phone to call the agriculture and nutrition counsellors to consult about any problem related to either the homestead garden or child’s feeding and health issues. A woman said,


*“I can call (my husband) in case of any problem using this mobile phone. I have been benefited as my husband has one mobile phone that he always keeps with him and carry wherever he goes. Now, if my husband goes outside he calls me in my phone if necessary, isn’t it good for me?”*
(Woman aged 31)

#### 3.1.8. Women’s Experiences on Cash Transfer Through ‘bKash’

All women reported receiving the cash through bKash mobile banking system. They also stated that this project supported them to open a bKash account through their mobile phones and informed them about the place and process of cash collection. However, having no national identity card, which is a requirement for registering for a bKash account, was a barrier to open an account for a few of the women. Husbands or a male family member supported the women to open the account in their name. They reported that they received a total of BDT 1200 (GBP 10) as the monthly cash transfer.

Although some of the women withdrew the money on their own, mostly husbands were primarily responsible for withdrawing the cash. The biggest barrier for them to withdraw the cash by themselves was their limited access to the marketplace, where the bKash agents are located.


*“They sent message (SMS) to inform to withdraw the money. I have withdrawn the money by myself. There was no problem to withdraw it. We, all who received the cash, have a specific bKash number that is used to withdraw cash from the bKash shop in the bazaar.”*
(Woman aged 22)


*“My husband has withdrawn the cash. There are many people in the market. I am not allowed to go to market.”*
(Woman aged 35)

#### 3.1.9. Women’s Decision Making on Spending the Cash

Most of the women said that they took the decision about when and what to spend the money on either on their own or jointly with their husbands. Some of them felt that they were allowed to take the decision because the money was given to them and targets their child’s welfare and better nutrition.


*“Everyone has good comments on my receiving money from this project. I can spend money as I wish, my husband says nothing. Now, I can buy anything that I prefer for my child. There is no need to ask money from my husband. I, myself, spend my own money.”*
(Woman aged 19)

#### 3.1.10. Perceptions of the Importance of Cash Transfer

The women maintained that cash transfer increased their affordability and income, while counselling generated awareness how to spend the cash effectively. As they became aware about the benefits of consumption of nutritious food, they preferred spending the cash to purchase foods for their family, especially animal source foods like egg and milk for their children. Some women spent the money on clothing, emergency health care such as doctor’s fees and medicine of the family members.

Similarly, the nutrition counsellor observed that providing a combination of nutrition messages and cash transfer was fruitful by complementing each other.


*“(The families) need advice, but quite often they can’t materialise advice they don’t have money. Now, if we give them money and no advice, they wouldn’t properly utilise the advice. Thus, if we provide them both advice and cash, they would be able to make proper use of the both. … If you give a poor family some meat, you also need to provide them some Takas so they can buy some oil and spices to cook and eat the meat.”*
(Agriculture Counsellor)

The family often spent the money to buy seeds and for fencing the garden as advised by the agriculture counsellor. Some relatively well-off women saved a portion of the money for emergencies and invest on income generating activities such as buying a hen to raise.

### 3.2. Quantitative Findings

#### 3.2.1. Characteristics of the Study Participants

Ninety percent of the (52/58) women participated in the survey were mothers of children aged 0–23 months (Table 2). Nearly the same proportion of the women was from the age groups of 15–24 years (48%, 28 out of 58) and 25–34 years (47%, 27 out of 58). Most of their husbands (96%, 56/58) were the main earning members of the family. The highest proportion of them was either unskilled labourer or skilled workers.

#### 3.2.2. Participation in Counselling on Homestead Gardening

As the Table 3 shows, all beneficiaries received the counselling from the agriculture counsellor and established their garden at their homestead. Almost all of them produced both leafy and non-leafy vegetables in their garden. A large percentage of them (86%) purchased seeds from local shops, while one in five of them had their own seed collections. Similarly, more than 90% of them purchased fertilizer from local shops and one in four of them produced compost at home. In response to the question about the sources of information beside the project agriculture counsellor, the respondents reported their husbands and their relatives as major sources of information on homestead gardening.

#### 3.2.3. Participation in Nutrition Counselling

All of the beneficiaries received nutrition counselling from the nutrition counsellors in this project (Table 4).

The majority of them (85%) reported receive counselling weekly, while less than one in five reported counselling occurring fortnightly. Most of them reported receiving information on complementary feeding (95%) and breastfeeding (88%). About 75% of them had information on personal hygiene, while one fifth of them mentioned about other information such as nutritious foods.

#### 3.2.4. Receipt of Cash Transfer through Mobile Banking

All of the participating women reported receiving a mobile phone from the project (Table 5).

Almost all of them used the mobile phone for making and receiving calls, while a few of them also used it for sending SMS (2%). All of the women received their cash through bKash. Although all of them were supposed to receive a cash payment of BDT 1200 (GBP 10) in total per month, about 22% of them also reported paying a very small service charge to the bKash agent when they collected the money. Most of the respondents spent the cash for buying foods (91%), around one fifth of them spent it on purchasing seeds or fertilizers. More than half of them (57%) also used the cash for other purposes such as health check-up, medicine, buying clothes for the children, and fencing the homestead garden.

## 4. Discussion

Our study revealed the intervention is a feasible and acceptable way of integrating nutrition education supported by a mobile app, face-to-face nutrition-sensitive agriculture counselling and support with unconditional cash transfer with target of improving maternal and child nutrition among the poor and nutrition insecure households in rural Bangladesh.

The agriculture workers counselling on the techniques of homestead gardening along with demonstrations can be effective to establish fruits and vegetables gardens in rural poor households, even though they have very little homestead land. The women from these households participated in the counselling with their husbands or other family members. It developed their capacity to build their own homestead gardens with the support of their family members. However, the major challenges related to gardening were the poor quality of seeds, heavy monsoon rains and damage of seed beds and the plants by hens, ducks and domestic cattle. Most women were able to solve these problems by taking initiatives as advised by the agriculture counsellor; for example, purchasing good quality seeds and fencing the garden beds. Women also demonstrated their interests by interacting with the nutrition counsellors who visited them. The nutrition counselling generated the women’s awareness on dietary diversity, and they preferred to consume vegetables produced in their gardens and use the cash to buy nutritious foods such as fruits and animal source foods (i.e. egg and milk), especially for their children. The participants were also satisfied with the timing of counselling, fortnightly group meetings and weekly home visits, as they participated only for half an hour usually in their leisure time. The study also revealed that the cash transfer through bKash mobile banking system is feasible to implement. The beneficiary families did not face any significant difficulties to collect the cash. However, a small fee for withdrawing the cash charged by the bKash agent posed a threat of developing mistrust among the beneficiaries. Cultural barriers to women’s mobility in crowded public places and not having a national identity card disabling them from opening an account appeared as barriers to cash transfer through mobile banking. But our study demonstrated husbands’ cooperation and support to overcome these barriers. bKash agents visiting the women at home to handover the cash can solve part of this challenge. Importantly, however, the cooperation of the husband further exhibits the potential of family involvement to such intervention to be implemented. Overall, the pilot intervention was feasible and widely accepted by the target audience.

Our findings will inform program implementers and policy makers to develop effective interventions that are tailored for the poor families in Bangladesh. Despite progress in reducing child undernutrition, there is still increasing inequality with less improvements in child nutrition in the poorest families in Bangladesh [8,23,24,25]. In the context of inequality, it is crucial to develop nutrition interventions targeting the poor. Our results suggested positive implications of integrating nutrition education, nutrition-sensitive agriculture counselling and support with unconditional cash transfer on a mobile platform to improve food security and dietary diversity for reducing maternal and child undernutrition among the poor in rural Bangladesh. Our approach is supported by the 2013 Lancet Series on Maternal and Child Nutrition that highlighted the potential benefit of combining nutrition-specific and nutrition-sensitive interventions during the first 1000 days from conception till the child is two years old to prevent maternal and child undernutrition [26,27].

Currently, Bangladesh has an extensive mobile phone network with about 129.6 million mobile phone users [28], which increases the potential to up-scale nutrition sensitive interventions using mobile platforms. However, evidence from a large survey in rural Bangladesh showed that only one third of the women owned a mobile phone and they were less likely to be an owner of a phone than the male members of the family [29]. Poorer women would even be less likely to own a mobile phone. Thus, our idea to provide mobile phones to the women in poor households as part of the social transfer creates an opportunity for them to use mobile banking system to receive social safety net payments. Moreover, it enhances their ability to access information and communicate with agriculture and nutrition workers for support, when necessary.

Our findings on spending cash for purchasing nutritious foods implies that unconditional cash transfer would lead to improved food security and dietary diversity, which is consistent with earlier cash transfer studies in LMICs [14,30,31]. A randomized controlled trial in Bangladesh revealed that cash transfer plus nutrition behaviour change communications improved the maternal awareness on iron deficiency and child’s intake of multiple micronutrients powder [32]. Strong evidence from a cluster randomized control trial in rural Bangladesh indicates that social transfers (cash or food) alone cannot prevent child stunting, but combining cash and nutrition BCC did improve the growth of children compared to a control group [33].

Our study presented a strong justification for using mobile phone apps to strengthen the counselling capacity of the community level agriculture and nutrition workers. This concept is supported by earlier evidence that reported the use of smartphone to strengthen the community health worker’s capacity to provide maternal, neonatal and child health services in Bangladesh [21]. A systematic review on mobile health interventions in LMICs suggested the mobile phone training apps are useful mHealth or mLearning tools, which could directly support community health workers with access to information and more effective communication with their clients [34]. Recently a study in South Asia emphasised on the community level workers’ capacity building on nutrition literacy and use of technology to improve the nutrition sensitive agriculture inventions in this region [35]. The training we provided to the community based nutrition and agriculture workers on counselling skills, nutrition information and use of mobile technology can fulfil the gap if it is delivered in a larger scale.

The strength of this study includes our mixed-methods approach. We also took account of both the beneficiaries and the project implementers to assess the feasibility of the intervention. The use of mixed-methods and multiple sources of data ensured both methodological and source triangulations of our findings. However, the study also has some limitation that need to be considered when interpreting our findings. The short duration of the intervention prevented us from examining the intervention in all crop seasons. We conducted the study with a specific group (the poor) of the community in a smaller setting. This might restrict the generalizability of our findings. However, generalizing the results was not the main objective of this study. Our findings should be tested in a larger scale to make them more generalizable.

## 5. Conclusions

The study provided evidence of feasibility and acceptability for combining nutrition education, nutrition-sensitive agriculture counselling and support with unconditional cash transfer targeted at improving maternal and child nutrition among the poor and nutrition insecure households in rural Bangladesh. The findings have relevance to other similar settings in low- and middle-income countries. The experience will be used to design a large-scale trial to test the impact of the intervention on diet quality for reducing maternal and child undernutrition in a resource poor setting in the near future. The primary outcome of the trial will be child stunting at 6 to 24 months. We would measure other key variables such as, women’s and children’s dietary diversity, change in breastfeeding and complementary feeding practices, food security, increase in home-gardening, diversity of home crop production, and some secondary variables such as change in women’s empowerment status.

## Figures and Tables

**Table 1 ijerph-17-04153-t001:** Mobile phone features, recipients, and their participation in data collection.

Recipient	Type of Mobile Phone	Features Embedded	N for Survey	N for Qual. Data
Women	Java	Monthly re-charge for talk time	58	20
Agriculture counsellor	Smartphone	Counselling app	2	2
Nutrition counselor	Smartphone	Counselling app	2	2

**Table 2 ijerph-17-04153-t002:** Characteristics of the women participated in this study (N = 58).

Women’s Characteristics	N = 58	
	*n*	%
Age of children in categories		
0–23 months	52	89.6
24–59 months	6	10.4
Age of women in categories		
15–24 years	28	48.3
25–34 years	27	46.6
35–44 years	03	5.2
Husband’s current working status		
Working	56	96.5
Not working	2	3.5
Husband’s Occupation		
Unskilled laborer	24	41.4
Skilled worker	24	41.4
Small Business/Trade	6	10.3
Service holder	2	3.4
Others	2	3.4

**Table 3 ijerph-17-04153-t003:** Feasibility of agriculture counselling on homestead gardening.

Agriculture Counselling	N = 58	
	n	%
Received counselling on homestead gardening from agriculture counsellor	58	100.0
Established homestead gardening after counselling	58	100.0
Vegetables produced in homestead gardening		
Leafy vegetables, green or colourful	58	100.0
Non-leafy vegetables, green or colourful	57	98.3
Source of seed collection		
Local shop	50	86.2
GO/NGO	7	12.1
Neighbour	5	8.6
Own collection	12	20.7
Source of fertilizer collection		
Local shop	53	91.4
GO/NGO	3	5.2
Neighbour	3	5.2
Home-made	15	25.9
Sources of information except agriculture counsellor		
Husband	42	72.4
Neighbour	9	15.5
NGOs	2	3.4
Relatives	25	43.1

**Table 4 ijerph-17-04153-t004:** Feasibility of nutrition counselling.

Nutrition Counselling	N = 58	
	n	%
Received nutrition counselling	58	100.0
Frequency of counselling		
fortnightly	9	15.5
weekly	49	84.5
Information received from counselling		
Breastfeeding	51	87.9
Complementary feeding	55	94.8
Personal Hygiene	43	74.1
Others (i.e. Nutritious foods)	11	19.0

**Table 5 ijerph-17-04153-t005:** Feasibility of cash transfer through mobile phones.

Cash Transfer through Mobile Phone	N = 58	
	n	%
Received mobile phones from project	58	100
Use of mobile phone		
Calling	57	98.3
Sending SMS	2	3.4
Listening to music	8	13.8
Watching videos	9	15.5
Received cash through Bkash	58	100.0
Person responsible for cash withdrawal		
Women herself	23	39.7
Husband	33	56.9
Children	58	100.0
Relatives	5	8.6
Amount of cash received, monthly		
1200 Taka only	45	77.6
1200 Taka plus service charge	13	22.4
Purpose of spending cash		
Purchase seeds	13	22.4
Purchase fertilizer	10	17.2
Purchase foods	53	91.4
Child’s study	3	5.2
Others *	34	58.6

* Other purposes include i.e. health, medicine, small livestock, clothes for the children, fencing etc.

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
