# Peer review of "In the past, the seeds I planted often didn’t grow.” A Mixed-Methods Feasibility Assessment of Integrating Agriculture and Nutrition Behaviour Change Interventions with Cash Transfers in Rural Bangladesh"

_ijerph, 2020, doi:10.3390/ijerph17114153_

Round 1

Reviewer 1 Report

Summary

This manuscript describes an intervention package, that combined support for the establishment of home-gardens, nutritional counseling, and an unconditional cash transfer, in a group of 60 mothers in rural Bangladesh. This mixed-methods project smartly designed the intervention with methods of data collection that reveal the strengths and weaknesses of the intervention. Overall, the intervention appeared to be highly successful, with a few issues (poor quality seeds, effects of heavy rains, the need to put up fences to protect the garden etc.) that can be addressed in the future. These "lessons learned" will be used to inform a larger-scale trial examining the effect of this type of dual intervention (gardening and nutrition) on dietary quality and maternal-child undernutrition in the future. Overall, the content of the paper is exciting, original, and of significance to individuals in science and policy. The content is presented in an accessible manner and all aspects of the project appear to be technically sound,. The analysis will undoubtedly be of interest to a wide readership. Below I provide a few broad suggestions and some minor edits.

Broad Comments

Great introduction laying out the problem of undernutrition in rural Bangladesh and providing a nuanced consideration of the connections between poverty, food insecurity, and hunger. I particularly appreciated the point that parallel interventions often lack integration and that agricultural interventions seldom have a nutritional component

That being said, the authors may reconsider opening the title of the paper with the quote, "In the past, the seeds I planted often didn't grow". While I see why this quote sums up the point of the paper - it is a bit confusing on the first read. My first time reading it,  I thought the paper might be about climate change. I suggest the authors think about the key words they used in the writing of the article and integrate those into the title instead.

As well, while the manuscript is generally well-written and balanced, there are a few sections that felt a bit short in relation to others. Specifically, the quantitative data analysis description (lines 192-93) is very brief. Maybe the authors can add in a bit more detail on what variables were being examined in this analysis? As well, Section 3.1.3 (lines 221-26) is short, perhaps there is another relevant quote?

Additionally, there are a few places where I was hoping to see a bit more information. For example, in line 251 a participant describes a video that combines nutrition, breastfeeding, hygiene etc., which is very exciting. Is there a link to this content for interested researchers? Or maybe more information about where these materials came from or how they were developed.

 This relates to a second question, how much does an intervention like this cost? How many people were involved in the execution of the project in total? Basically, I am looking for a few more details on the logistics of launching a project like this.

Finally, in terms of costs and benefits, was there thought about collecting biological (anthropometric measures etc.) to examine long-term child health in response to the intervention? On line 415 the authors mention that there is "evidence from a cluster randomized control trial in rural Bangladesh indicates that social transfers 416 (cash or food) alone cannot prevent child stunting, but combining cash and nutrition BCC did 417 improve the growth of children compared to a control group (33)" and also say the results of this study will be used in a larger trial in the future. Just a few words about what outcomes would be measured in the future would be useful. As well, the authors conclude on 390 that this work is relevant in Bangladesh, but I would argue it has global relevance.

Specific/Minor Comments

Line 100 - sentence beginning with "we conducted formative research" - is awkwardly worded

Line 124 - recharge --> charger

Line 167 - agriculture worker --> agricultural worker?

Line 170 - "the challenges and facilitators encountered as the project" --> "that facilitators"

Line 190-191 - awkwardly worded

Line 220 - women --> woman

Author Response

Thank you for your constructive feedback. We have addressed point-by-point the comments and explained the revision made in response to the comments. We have highlighted the revisions to manuscript using Track Changes.

Reviewer 1

  1. Comment: The authors may reconsider opening the title of the paper with the quote, "In the past, the seeds I planted often didn’t grow". . . . I suggest the authors think about the key words they used in the writing of the article and integrate those into the title instead.

Response: We thank the reviewer for the suggestion. We have discarded the quote from the title. The revised title is, “A mixed-methods feasibility assessment of integrating agriculture and nutrition behaviour change interventions with cash transfers in rural Bangladesh.”

  1. Comment: As well, while the manuscript is generally well-written and balanced, there are a few sections that felt a bit short in relation to others. Specifically, the quantitative data analysis description (lines 192-93) is very brief. Maybe the authors can add in a bit more detail on what variables were being examined in this analysis?

Response: We have expanded the quantitative data analysis. Now the revised manuscript contains the flowing text in lines 215-218.

“We assessed the participants’ socio-demographic characteristics, participation in both nutrition and home gardening counselling, types of information received, receipt of cash transfers, use of the cash, and use of mobile phone by women, in the qualitative analysis to complement the qualitative data.”

  1. Comments: As well, Section 3.1.3 (lines 221-26) is short, perhaps there is another relevant quote?

Response: We added the following a quote on agriculture knowledge retention in the revised paper in lines 252-256.

“They (Agriculture Counsellor) advised us to plough the land, apply cow dung, faeces of chicken and goat etc, and urea and patash (potassium) fertilizers to make a bed to plant red amaranth, napa leaves and spinach. They asked us to make separate beds; if we make beds then excess water will drain. … To grow bottle gourd, they told us to dig [soil] one arm deep, apply cow dung and goat manure, keep it like that for 5/6 days and then plant gourd seeds. (Woman aged 25)

  1. Comment: Additionally, there are a few places where I was hoping to see a bit more information. For example, in line 251 a participant describes a video that combines nutrition, breastfeeding, hygiene etc., which is very exciting. Is there a link to this content for interested researchers? Or maybe more information about where these materials came from or how they were developed.

Response: We thank the author for the suggestion. While it is a challenge to provide more information and keep the length of the paper to an acceptable limit at the same time, we have added a brief description of the intervention material development process on page three. We have also attached some of the messages as an additional material. The following text has been added in lines 122 and 132.

“. . . The formative research provided us insight into the key elements to be included in the education messages. Nutrition and agriculture experts among the authors developed the messages that were shared with professionals from public NGO sectors for feedback. We field-tested the final messages to check for understandability among the beneficiary community. A skilled programmer experienced in CommCare mobile phone applications, developed an app about these nutrition and agriculture messages that was embedded in the smart phones to be used by the trained counsellors for counselling the participating women. The app also included a resource library that accumulated pictorial illustrations and videos about appropriate infant and feeding and home gardening practices. The counsellors showed the video or illustration linked to the specific messages if a woman asked for an elaboration of the messages.”

  1. Comment: This relates to a second question, how much does an intervention like this cost? How many people were involved in the execution of the project in total? Basically, I am looking for a few more details on the logistics of launching a project like this.

Response: We used the exiting infra-structure of our local project partner, a local NGO working in the district. A group of their staff with relevant experience was involved in the project in addition to their other assignments. One project officer, whose salary was partially incurred from this project oversaw the field activities. We paid two nutrition counsellors and two agriculture counsellors full-time for six months. We also involved a partially paid project coordinator, who was based at an organization based in the capital city to manage the project and maintain collaboration with the University of Sydney (the grant recipient).      

  1. Comment: Finally, in terms of costs and benefits, was there thought about collecting biological (anthropometric measures etc.) to examine long-term child health in response to the intervention?

Response: We did not aim for collecting anthropometric measurement data in this pilot study. The intervention lasted for only six months and we anticipated no significantly measurable change in anthropometry within such a short duration. Future interventions that might use the feasibility lessons of this study would measure for a wider range of nutrition indicators.  

  1. Comment: On line 415 the authors mention that there is "evidence from a cluster randomized control trial in rural Bangladesh indicates that social transfers (cash or food) alone cannot prevent child stunting, but combining cash and nutrition BCC did improve the growth of children compared to a control group (33)" and also say the results of this study will be used in a larger trial in the future. Just a few words about what outcomes would be measured in the future would be useful.

Response: The text below has been added to the revised paper in lines 476-480 to address the comment.

“In the future trial, our primary outcome will be child stunting at 6 to 24 months. We would measure other key variables such as, women’s and children’s dietary diversity, change in breastfeeding and complementary feeding practices, food security, increase in home-gardening, diversity of home crop production, and some secondary variables such as change in women’s empowerment status.”

  1. Comment: As well, the authors conclude on that this work is relevant in Bangladesh, but I would argue it has global relevance.

Response: We appreciated the comment. We have inserted the sentence below to the concluding paragraph in lines 473-474.

“The findings have relevance to other similar settings in low- and middle-income countries.”

  1. Specific/Minor Comments

Line 100 - sentence beginning with "we conducted formative research" - is awkwardly worded

Line 124 - recharge --> charger

Line 167 - agriculture worker --> agricultural worker?

Line 170 - "the challenges and facilitators encountered as the project" --> "that facilitators"

Line 190-191 - awkwardly worded

Line 220 - women --> woman

Response: We have now corrected the errors identified by the reviewer.

Reviewer 2 Report

Feasibility and acceptability of an  intervention package that combines nutrition counselling, counselling and support for home-gardening, and unconditional cash transfers delivered to women on a mobile platform for  improving maternal and child nutrition behaviours among low-income families in rural Bangladesh was assessed.  A mixed methods approach identified challenges and solutions. Usefulness of mobile technologies was assessed.

This interesting and informative paper should interest a wide audience.  There are some omissions and the flow could be improved – I had to go backwards and forwards to understand better.   It does require rewriting and sequencing so that the aims, the method and the results flow better and are more explicit with the use of the same words. e.g. compliance and also identifying the messages used - if too extensive this should be supplementary material - are there any the authors would not use again or would prioritise for delivery?

Major

There is no apparent reference to how ethics approval was obtained and how consent was provided by participants including key informants.

The study has three aims…1. feasibility, acceptance and compliance 2. Participants understanding knowledge retention and perceived usefulness 3 expenditure pattern  - the abstract should address these more rigorously. See also minor comments.

Two study sites were selected – one close to the Tista river – please explain the rationale for this distinction and how many of the 60 households were from each site?  Are these sites within the Kurigram area?

It seems that the mobile technology was nutrition and agricultural messages embedded in the smart phones used by trained counsellors. In addition mobile phones were provided to 60 women with a cash transfer application embedded. Further project agriculture workers were provided with a smartphone.  A figure showing the number of cell and smart phones issued and to who. with what uses would be useful. Perhaps this figure could also include the number of people involved at each step e.g. 60 households but 20 indepth interviews. Key informants – how many?

Nutrition and agricultural messages – could these messages be provided in a table – how many were there in each domain and what was the focus.  Breastfeeding could be a third domain? Do the authors have a measure of which messages appeared to have the most impact? One quote – line 264 mentions vitamins which are a complex construct – how was this message conveyed? How many of these women were literate? It is said that only a few could read or write an SMS line 269.

Minor

In the abstract please state the number of women, family members and project staff that completed the cross sectional survey

Compliance – how was this measured – is this the 58/60 who completed the questionnaire?

Line 157 – the last sentence of this paragraph does not make sense – should the ‘of’ be ‘or’

Author Response

Thank you for your constructive feedback. We have addressed point-by-point the comments and explained the revision made in response to the comments. We have highlighted the revisions to manuscript using Track Changes.

Reviewer 2

  1. Comment: There is no apparent reference to how ethics approval was obtained and how consent was provided by participants including key informants.

Response: We thank the reviewer for pointing out this missed information. We submitted the ethics approval document to the journal during the submission. We have added a paragraph on ethics to the revised paper with the following text in lines 190- 195.

“The study was approved by the Ethical Review Committee of the International Centre for Diarrhoeal Disease Research, Bangladesh (icddr,b). We obtained written informed consent from each participant of the survey, in-interviews and key-informant interviews. Verbal audio recorded consent was also obtained for audio recording of the interviews. The participants were assured that their participation in the data collection was completely voluntary. We also assured anonymity and privacy of information they would provide.”

  1. Comment: The study has three aims…1. feasibility, acceptance and compliance 2. Participants understanding knowledge retention and perceived usefulness 3 expenditure pattern  - the abstract should address these more rigorously. See also minor comments.

Response: The abstract has been modified to reflect the comment as much as possible keeping the prescribed word limit. We have rearranged the abstract to more clearly present information about acceptance, compliance, and feasibility between lines 27 and 35. The abstract now reads as follows:

“Combining agriculture with behaviour change communication and other nutrition-sensitive interventions could improve feeding practices to reduce maternal and child undernutrition. Such integrated intervention requires rigorous design and an appropriate implementation strategy to generate an impact. We assessed feasibility and acceptability of an intervention package that combines nutrition counselling, counselling and support for home-gardening, and unconditional cash transfers delivered to women on a mobile platform for improving maternal and child nutrition behaviours among low-income families in rural Bangladesh. We used mixed-methods including in-depth interviews with women (20), key-informant interviews with project workers (6), and a cross sectional survey of women (60). Women well-accepted the intervention and reported to be benefited by acquiring new skills and information on home gardening and nutrition. They established homestead gardens of seasonal vegetables successfully and were able to find a solution for major challenges. All women received the cash transfer. Ninety-one percent of women spent the cash for buying foods, 20% spent it on purchasing seeds or fertilizers and 57% used it for medical and livelihood purchases. Project staff and mobile banking agent reported no difficulty in cash transfer. Combining nutrition-specific and -sensitive interventions is a feasible and acceptable approach. Using mobile technologies can provide additional benefits for the intervention to reach the disadvantage families in rural settings.”

  1. Comment: Two study sites were selected – one close to the Tista river – please explain the rationale for this distinction and how many of the 60 households were from each site?  Are these sites within the Kurigram area?

Response: The villages and settlements in the area are relatively small comparing to many other areas in the country. We had to select households from two villages to recruit sixty women with a child under five years of age. One village was a little away from the river but had similar socio-economic conditions. Both sites are within Kerrigan with about one kilometer apart. Additional information has been inserted to the paragraph in lines 95-98.  

  1. Comment: It seems that the mobile technology was nutrition and agricultural messages embedded in the smart phones used by trained counsellors. In addition mobile phones were provided to 60 women with a cash transfer application embedded. Further project agriculture workers were provided with a smartphone.  A figure showing the number of cell and smart phones issued and to who. with what uses would be useful. Perhaps this figure could also include the number of people involved at each step e.g. 60 households but 20 indepth interviews. Key informants – how many?

Response: We have added a table as suggested by the reviewer in page 4. We have also put the number of key-informants (six) in the abstract, and methods section in lines 181-184.

Table 1. Mobile phone features, recipients and their participation in data collection.

Recipient

Type of mobile phone

Features embedded

N for Survey

N for qual. data

Women

Java

Monthly re-charge for talk time

58

20

Agriculture counsellor

Smartphone

Counselling app

2

2

Nutrition counselor

Smartphone

Counselling app

2

2

  1. Comment: Nutrition and agricultural messages – could these messages be provided in a table – how many were there in each domain and what was the focus.  Breastfeeding could be a third domain? Do the authors have a measure of which messages appeared to have the most impact? One quote – line 264 mentions vitamins which are a complex construct – how was this message conveyed? How many of these women were literate? It is said that only a few could read or write an SMS line 269.

Response: Please see our response to the comment number 4 of the reviewer 1. We have attached the summary of nutrition and agriculture messages as an additional material. Breastfeeding is a part of the nutrition messages.

We haven’t assessed the impact of any message because assessing an impact is beyond the aim of this feasibility study. Moreover, attempt to measure the impact of counselling provided by a very short duration (six months) pilot intervention mostly unlikely to provide any results. We assumed six months was too short to generate an impact, hence our aim was to assess the feasibility of the intervention. A future larger study with appropriate design would aim of measuring the impact of the intervention. 

The women’s literacy in the study site is low comparing to most of the other regions in Bangladesh. This was the reason for our plan not to send pushed SMS to the women. Rather, our counsellors used pictorial, video and written messages on the app and explained the messages to the women as they counselled them. Out of the twenty who were interviewed for qualitative data, six were literate.

  1. Comment: In the abstract please state the number of women, family members and project staff that completed the cross sectional survey.

Response: The cross-sectional quantitative survey was conducted with women. The family members and project along with a sub-sample of women participated in the qualitative survey. We have adjusted the information in the revised abstract.

  1. Comment: Compliance – how was this measured – is this the 58/60 who completed the questionnaire?

Response:  Yes.

  1. Comment: Line 157 – the last sentence of this paragraph does not make sense – should the ‘of’ be ‘or’.

Response: This typo has been corrected.

Round 2

Reviewer 1 Report

The authors have adequately addressed the reviewer comments and, from my perspective, the manuscript is ready for publication. 

Reviewer 2 Report

The responses you have provided add some clarity. Thank you 

However I am still confused by the numbers - in the abstract it is 60 women but in the tables the number is 58. Similarly in the abstract 6 project workers are listed but 4 - 2 agriculture and 2 nutrition counsellor are in table 1. 

The English requires extensive editing - too numerous to comment individually. 

e.g. the first sentence Combining agriculture with behaviour change communication.. does not scan.  Agriculture is a noun - but what aspect of agriculture is referred too. 

(icddr,b)  should be capitals

The authors state in their response to reviewer 2 that they have provided additional material but this is not evident in the revised article

Please see our response to the comment number 4 of the reviewer 1. We have attached the summary of nutrition and agriculture messages as an additional material. Breastfeeding is a part of the nutrition messages.